depression; anxiety; network analysis; LMIC; India

**Corresponding author:**
Cemile Ceren Sönmez;
Email: ccs2146@tc.columbia.edu

# Symptom networks of common mental disorders in public versus private healthcare settings in India

Cemile Ceren Sönmez[1,2] ⓘ, Helen Verdeli[1], Matteo Malgaroli[3],
Jaime Delgadillo[4] and Bryan Keller[5]

[1]Counseling and Clinical Psychology Department, Teachers College, Columbia University, New York, USA; [2]Institute for Global Health, University College London, London, UK; [3]Department of Psychiatry, NYU Grossman School of Medicine, New York, USA; [4]Department of Psychology, The University of Sheffield, Sheffield, UK and [5]Department of Human Development, Teachers College, Columbia University, New York, USA

## Abstract

We present a series of network analyses aiming to uncover the symptom constellations of depression, anxiety and somatization among 2,796 adult primary health care attendees in Goa, India, a low- and middle-income country (LMIC). Depression and anxiety are the leading neuropsychiatric causes of disability. Yet, the diagnostic boundaries and the characteristics of their dynamically intertwined symptom constellations remain obscure, particularly in non-Western settings. Regularized partial correlation networks were estimated and the diagnostic boundaries were explored using community detection analysis. The global and local connectivity of network structures of public versus private healthcare settings and treatment responders versus nonresponders were compared with a permutation test. Overall, depressed mood, panic, fatigue, concentration problems and somatic symptoms were the most central. Leveraging the longitudinal nature of the data, our analyses revealed baseline networks did not differ across treatment responders and nonresponders. The results did not support distinct illness subclusters of the CMDs. For public healthcare settings, panic was the most central symptom, whereas in private, fatigue was the most central. Findings highlight varying mechanism of illness development across socioeconomic backgrounds, with potential implications for case identification and treatment. This is the first study directly comparing the symptom constellations of two socioeconomically different groups in an LMIC.

## Impact statement

Depressive disorders, along with anxiety and somatic pain, are among the top leading causes of non-fatal disease burden globally. In 2019, depression was the top leading cause of burden of disease for countries that are at the lower end of socioeconomic development. Especially in low-resource settings, the identification and treatment of these illnesses pose a grand challenge. Although depression, anxiety, and somatic symptoms are highly prevalent and debilitating, the diagnostic boundaries and the mechanism of development of these illnesses are not well understood. One reason for this might be that these illnesses are often comorbid and present a heterogeneous clinical picture. The complex and dynamic relationships between symptoms call for a nondiagnostic and dynamic modeling technique. In this article, we used the "network approach" to map out the symptoms of depression, anxiety and somatic drawn from a sample of adult primary care attendees in India. Our work addressed several methodological weaknesses of the symptom network literature by using the composite subscale scores of a culturally valid clinical interview with no skip algorithm or overlapping variables. We found "panic" symptom, conceptualized as "intense anxiety/nervousness" or "tension" to be the most central in public healthcare settings, while "fatigue" was the most central in private healthcare settings. This indicates some kind of stress/threat response might be the hallmark of common mental disorders among those who are the most economically disadvantaged in India, and potentially in the region. Studying the complexity of the symptom-to-symptom relationships for these highly comorbid conditions can help flag and target the key symptoms that sit at the core of the illness, hence allowing for the optimal use of the limited resources. To our knowledge, this is the first study comparing the network structure of common mental disorders of primary care patients from different socioeconomic backgrounds in an low-income country.



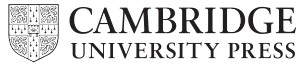

## Introduction

Globally, mental disorders are the second leading cause of years lived with disability. Depressive and anxiety disorders together account for more than 60% of the disability-adjusted life years

(DALYs) for mental disorders (GBD 2019 Mental Disorders Collaborators 2022). Depression, anxiety and somatic symptom disorders have high comorbidity rates ranging from 40% to 80% (Kessler et al. 2005; Lamers et al. 2011), share similar biological markers (Goodkind et al. 2015; Drysdale et al. 2017) and psychological vulnerabilities (Brown and Barlow 2009), and respond similarly to psychotherapy and pharmacotherapy (Cuijpers et al. 2013). Thus, the diagnostic criteria and the illness development are not well understood, particularly in non-Western settings. The present study aimed to uncover the symptom constellation and the illness comorbidity in a primary care sample in India, using network analysis.

Our current conceptualization of mental illness, the common cause model (Kendler et al. 2011), which assumes that a latent factor causes all symptoms of a disorder, has been criticized due to its conceptual, statistical and clinical limitations. A body of research challenges the idea of distinct disease categories particularly for depression and anxiety (Kessler et al. 2005; Sullivan et al. 2013; Goodkind et al. 2015; Drysdale et al. 2017). The symptom network approach was proposed as an alternative, where psychiatric disorders are thought to consist of a constellation of symptoms connected to each other through a dynamic and mutually reinforcing network (Cramer et al. 2010). Based on this, a direct link between two symptoms (e.g., lack of sleep and somatic symptoms) is assumed to exist outside of what could be explained by an underlying factor. Symptoms that are "central," meaning highly connected to the rest of the symptoms in the network, could inform us about the state of the entire network and potentially serve as good therapy targets (e.g., Fried and Cramer 2017).

### Depression and anxiety symptom networks globally

Growing evidence supports both the statistical appropriateness and clinical usefulness of network models (Cramer et al. 2012; Fried et al. 2015). In a systematic review, alongside depressed mood, fatigue was frequently reported as another central symptom (Malgaroli et al. 2021). Networks of depression have been explored cross-culturally, and in Asian cultures. Recent research with depressed adults from various Asian countries reported sad mood (Park et al. 2020; Wasil et al. 2020) and fatigue (Garabiles et al. 2019) as central symptoms of depression. Different from most findings in Western countries, feeling like a failure was also reported as one of the most central depressive symptoms among adolescents in India (Wasil et al. 2020) and Han Chinese women (Kendler et al. 2018).

Anxiety symptoms were frequently investigated along with depressive symptoms. The first empirical paper on symptom networks of psychopathology examined depression and anxiety symptoms in a national survey from the United States (Cramer et al. 2010). Subsequent studies involving international datasets and samples from Asian cultures highlighted anxiety as a central symptom in depressive symptomatology. In an international study, the item "I was close to panic" was the most central among 21 items drawn from the Depression, Anxiety and Stress Scale (Van den Bergh et al. 2021). The authors highlighted that one nationality, Malaysian was overrepresented in this sample. Two other studies reported anxiety as a central symptom among depressed adults from South Korea. In one, Park and Kim 2020 used the Beck Depression Inventory (BDI) and Beck Anxiety Inventory (BAI) and found anxiety symptoms (i.e., lightheadedness, feeling of choking and feeling scared) to be as central as depressive symptoms. In a subsequent nationwide study, Park et al. 2021 reported psychic

anxiety to be the most central symptom of the 17-item Hamilton Depression Scale (HAM-D). Overall, while sad mood, lack of interest and fatigue were central depressive symptoms across cultures, in some international studies, anxiety symptoms were found either as central as (Park and Kim 2020) or more central than depressive symptoms (Park et al. 2021; Van den Bergh et al. 2021).

A major methodological limitation in the network literature concerns the assessment of psychological symptoms. The comorbidity networks are typically derived from self-report questionnaires (e.g., PHQ, BDI) or structured interviews (e.g., SCID, MINI; (Malgaroli et al. 2021) mirroring the diagnostic criteria for very specific disorders, such as Major Depressive Disorder and Generalized Anxiety Disorder. While exploring symptom dynamics based on diagnostic categories can be informative, this approach has several limitations. First, symptoms that belong to comorbid conditions (e.g., panic, somatic-related disorders) might be left out limiting our understanding of mental illness to already existing categories. Second, when exploring comorbidity networks, topologically similar symptoms might be included more than once (implicated in both conditions, e.g., sleep disturbance, fatigue). Third, in the case of self-report questionnaires, measurement problems may arise when one symptom domain (e.g., appetite) is assessed via opposite (e.g., diminished and increased appetite) or nested items (e.g., loss of appetite and weight loss) which may lead to biased centrality estimates (Fried and Cramer 2017). Fourth, the use of single items as nodes could increase measurement error (Fried and Cramer 2017). Fifth, the "skip-out" items (items that are skipped when a core symptom is not endorsed) embedded in many structured interviews may lead to overstated symptom correlations (Hoffman et al. 2019).

The present study addresses these methodological issues in several ways. First, data are drawn from the Revised Clinical Interview Schedule (CIS-R; Lewis et al. 1992) which was developed with an aim to assess common mental disorders (CMDs; Goldberg and Huxley 1992) as one aggregate category capturing all depression, anxiety-related and somatic symptoms listed in the Diagnostic and Statistical Manual (DSM; American Psychiatric Association 1987) and ICD World Health Organization 2004). The CIS-R is not structured around diagnostic categories. It assesses each symptom only once via multiple items and generates a composite subscale score for each domain (e.g., somatic subscale score). As a result, it neither has a skip logic nor overlapping symptoms. Furthermore, since the CIS-R has been locally validated and widely used in India, locally relevant symptoms such as irritability are also assessed (Andrew et al. 2012; Weaver 2017). Overall, the CIS-R offers an optimal ground to construct a culturally valid, comprehensive yet parsimonious symptom network.

### The present study

The present study consists of secondary analysis of data collected from primary care patients in Goa, India, as part of MANAS (Patel et al. 2010), a clinical trial aimed at testing the effectiveness of a collaborative stepped-care intervention led by lay health counselors (LHCs). In the MANAS trial, researchers included all primary care patients who screened positive on the 12-item General Health Questionnaire (GHQ-12). The present study has four aims.

*Aim 1.* In India, the world's most populous low-income country, the health system is heavily privatized and healthcare expenditures are a leading cause of poverty (Reddy et al. 2011). High rates of depression, anxiety and somatic symptoms are reported in primary care, ranging between 18.8% and 46% (Sen 1987; Patel et al. 1998, 2011).

Yet, little is known about the onset and mechanism of CMDs in adults, and the interplay between the symptoms. The first aim of this current study is to uncover the symptom network of CMDs in an adult primary care patient population in Goa.

*Aim 2.* Previous symptom network studies found distinct clusters of anxiety and depression with high intercluster connectivity (Park and Kim 2020; O'Driscoll et al. 2021; Van den Bergh et al. 2021). In India, factor analytic studies revealed that depression, anxiety and somatic phenomena are not clearly separated among primary care patients. Thus, the second aim is to investigate whether there are distinct communities of illness under the common mental disorder category using network analysis.

*Aim 3.* Mechanism of illness development and central symptoms may vary across different socioeconomic levels. One study descriptively compared the symptom networks of patients from countries with different income levels. Park and Kim 2020 found guilt, fatigue and suicidality to be more central in high-income countries (Hong Kong, Japan, Korea, Singapore and Taiwan) and persistent sadness, fatigue and loss of interest most central in middle-income countries (China, India, Indonesia, Malaysia and Thailand). The authors explained that the differences might be better explained by cultural elements since high-income countries were largely East Asian, whereas middle-income countries were largely South or Southeast Asian. To our knowledge, no study compared networks across different socioeconomic levels within the same culture. Thus, our third aim was to test for the differences in the network structures across different levels of socioeconomic background in India. Earlier findings from the MANAS trial revealed differential effectiveness of the intervention across public and private healthcare centers (Patel et al. 2010), potentially indicating public versus private settings might serve as an appropriate proxy for socioeconomic background. Therefore, we used the hospital setting as a grouping variable for this study.

*Aim 4.* Network density, meaning the overall strength of symptom connection, has also been examined in the context of prognosis and treatment response. In one study, those with persistent depression at 2-year follow-up had "tighter" meaning more connected networks at baseline, compared to remitters (Van Borkulo et al. 2015). Other studies found no such differences in the context of treatment (Schweren et al. 2018; O'Driscoll et al. 2021). Thus, the fourth aim of the present study is to compare the symptom network density of treatment responders at 2, 6 and 12-month follow-up, versus nonresponders.

## Methods

### Participants

The current study draws on baseline and follow-up data from a cluster randomized controlled trial (MANAS trial; Patel et al. 2010) testing the effectiveness of an LHC led collaborative stepped-care intervention. Eligible participants who screened positive for a CMD using the 12-item General Health Questionnaire ($n$ = 2,796) were included in the trial. Only baseline data are used to construct the symptom networks for the present study. We used follow-up data collected at 2, 6, and 12 months to identify treatment responders versus nonresponders.

### Measures

The CIS-R has adequate internal consistency ($\alpha$ = 0.82; Lewis et al., 1992) and was previously adapted and extensively used in Goa,

India (Patel et al. 1998, 2003). The measure assesses the presence and severity (duration, intensity, and frequency in the past week or month) of 12 nonpsychotic psychiatric symptoms, each captured through multiple-item questions. The following subscales were used in this study; somatic, fatigue, depressed mood, anxiety, worry, phobia, panic, irritability, sleep problems, worry about health, concentration problems and depressive ideas (e.g., hopelessness, suicidal thoughts). While worry subscale captured general worries about things and circumstances, anxiety subscale included items about anxious feelings, nervousness or tension. Depressive ideas subscale captured diurnal variation, restlessness, psychomotor agitation, feeling guilty, worthlessness, hopelessness and suicidal ideas.

Each subscale consisted of the sum of four or five binary items. The symptom subscale scores ranged from 0 to 4, except for fatigue and the depressive ideas ranging from 0 to 5. Two additional composite scores (i.e., changes in weight/appetite and functional impairment) were computed for the present study. The appetite and weight change ranged from 0 to 2 and functional impairment variable ranged from 0 to 3. The obsessions and compulsions subscales of the CIS-R have not been part of the interview used in MANAS trial, thus were not included in the analyses. See supplementary material for more details on the composition of individual variables.

### Statistical analysis

#### Partial correlation networks (Aim 1)

A network of partial correlations between symptoms (i.e., nodes) was estimated. The partial correlation coefficient (i.e., edge weight) between two focal nodes represents the strength of the linear relationship between them after conditioning on (i.e., partialing out) other nodes in the network (Epskamp and Fried 2018). Partial correlations, as opposed to marginal correlations (i.e., unconditional correlations), are more appropriate for network modeling because under some assumptions they provide information about possible causal relationships. A widely used method to investigate the importance of nodes is called centrality. There are three common centrality indices used: (1) strength refers to the sum of the weights of edges that are connected to a node, (2) closeness refers to the average distance from that node to all other nodes and (3) betweenness refers to the number of times a node is on the shortest path between two other nodes (Epskamp et al. 2018).

The accuracy and stability of the network structures were evaluated in three domains (Epskamp et al. 2018). First, the centrality stability (i.e., correlation stability, CS coefficient) was evaluated. This indicates the maximum proportion of cases that can be dropped to maintain the correlation between the original centrality indices using a case-dropping bootstrap and is recommended to be above 0.5 (Epskamp et al. 2018). Second, the edge-weight accuracy and stability was assessed through a nonparametric bootstrap using the bootnet R package (Epskamp et al. 2018). Bootnet generates plots showing the bootstrapped CIs of edge weights, and generally, smaller CIs indicate more accurate edge weights. Also shown on the plots, if the number of times an edge was estimated to be nonzero is high, the stability of the edge weights is considered to be high (Epskamp and Fried 2018). Third, centrality and edge-weight differences are tested with a bootstrap significance test. A bootstrapped confidence interval (CI) is constructed around the difference scores. If the CI overlaps with zero, the centrality of two nodes (or edge weights) is considered to not differ significantly (Epskamp et al. 2018).

Network models were estimated with the R package qgraph using the least absolute shrinkage and selection operator (LASSO; Tibshirani 2011) limiting the number of spurious edges. The lasso tuning parameter controlling the level of network sparsity was selected by minimizing the Extended Bayesian Information Criterion (EBIC; Chen and Chen 2008). The EBIC uses a hyperparameter which is set by the researcher typically between 0 and 0.5 (Foygel and Drton 2010), with higher values indicating more parsimonious models (i.e., fewer edges). For this study, the EBIC hyperparameter was set to 0.25 based on (Hevey 2018) recommendation for exploratory research.

Partial correlation networks, also called Gaussian graphical models (GGMs), assume nodes are normally distributed. Because node distributions were skewed for each variable, the source distribution was transformed into a target standard normal distribution (Epskamp and Fried 2018). A nonparametric transformation was used where intermediary cumulative distributions were utilized to create a bijective map between source and target distributions. This was implemented in R using huge package (Zhao et al. 2012).

As an alternative solution to skewed data, symptom networks were also estimated using Ising modeling using the IsingFit package in R treating all variables as binary (Epskamp et al. 2018), as well as the mixed graphical method (MGM) using the mgm package in R (Haslbeck and Waldorp 2020) where only highly skewed variables were treated as binary; namely, anxiety, panic, phobias, appetite and functional impairment. Items were dichotomized based on clinical significance: for anxiety, panic, and phobias, scores equal or greater than 2 were coded as "1," as this has been a typical approach (Jacob et al., 1998). For appetite and functioning, any changes from baseline were coded as "1." Only GGM results are presented due to superior stability and accuracy (Epskamp et al. 2018), see supplementary material for details regarding the MGM and Ising models.

### Comorbidity networks (Aim 2)

The community structure was assessed using the walktrap random walk algorithm (Pons and Latapy 2006) within the igraph package (Csardi and Nepusz 2006). The algorithm quantifies the quality of a partition with a measure of modularity. Positive modularity indicated a potential community structure, with higher values of modularity indicating better partitioning. Networks with strong community structures were shown to have modularity indices ranging between 0.3 and 0.7 (Newman and Girvan 2004).

### Group comparisons (aims 3 and 4)

The challenge when constructing a test of network invariance across groups is that the probability distributions for summary statistics for networks are not analytically tractable. An alternative is to test network invariance using a permutation test (Van Borkulo et al. 2015). Permutation testing was carried out in package NetworkComparisonTest in R (Borkulo et al. 2022) to compare the network structure of treatment responders versus nonresponders and public versus private healthcare settings. The comparison is done in three ways: (1) the global strength, meaning the sum of all edge weights of permuted data, (2) maximum difference in edge weights and (3) the Holm-Bonferroni corrected p-values per edge from the permutation test concerning differences in edge weight. For the group comparison that pertained to treatment response, a patient was a "responder" if they (1) were a CMD case at baseline, and (2) responded to treatment at 2 months of follow-up and (3) sustained their response over the 6- and 12-months follow-ups. To form a group of similar size, a "nonresponder" had to be a

CMD case at baseline and satisfy any of the following conditions: (1) maintain the status of the CMD case in all follow-up evaluations, (2) delayed response at 6 or 12 months follow-up or (3) relapse at 6 or 12 months of follow-up after responding at 2 months of follow-up. Since the network structure comparison based on treatment response might differ across treatment arms, these analyses were repeated separately for the intervention and control arms.

## Results

### Demographics

The total number of participants was 2,796, with 1,648 from 12 public healthcare facilities (58.9%) and 1,148 (41.1%) from 12 private general practitioner facilities. The sample was predominantly female (82%), with a mean age of 46.29 (SD = 13.12), and the mean years of education was 3.67 (SD = 4.14). Based on the CIS-R classification of caseness, a total of 2,242 (81%) met the criteria for a CMD. Among those, 1,032 (46%) had mixed anxiety-depressive disorder, 774 (35%) had depression and the remaining 436 (19%) had a pure anxiety disorder. A demographic breakdown and rates of CMD and MDD are shown in Table 1.

### Network accuracy and stability

The resulting network is presented in Figure 1. The case-dropping bootstrap revealed that the order of node strength is interpretable, with a CS coefficient CS(cor = 0.7) equal to 0.75 meaning the average correlation between the original and bootstrapped indices remains higher than 0.75 even when more than 30% of the cases are dropped. On the other hand, the CS coefficient indicated that closeness (CS (cor = 0.7) = 0.36) and betweenness (CS(cor = 0.7) = 0.28) remained lower than recommended threshold. Thus, node strength was chosen as the primary centrality index (see supplementary material, Figure 1).

### Symptom centrality and edge weights (Aim 1)

The GGM revealed that the symptom depressed mood had the highest centrality score, though it was not significantly different than centrality scores of the three following symptoms listed in decreasing order; panic, fatigue and concentration problems (Figure 2). The centrality of the somatic symptom was similar to panic, fatigue and concentration problems but it was significantly smaller than depressed mood. Sleep problems had the next highest centrality score, followed by anxiety, worry about health, phobias, worry and functional impairment, in decreasing order. Irritability and appetite problems followed next, with depressive ideas (capturing suicidality and hopelessness) being the least central symptom. The edges phobia-panic, anxiety-panic, and somatic-fatigue were reliably the three strongest since their bootstrapped CIs had no overlap with the CIs of the remaining edges. Visual inspection of the edge differences table revealed several other edges that are significantly stronger than most of the rest: depressed mood-worry, fatigue-concentration, concentration-depressed mood, functional impairment-concentration and worry-anxiety. The edge anxiety-phobias was significantly smaller than the rest; however, this edge was estimated nonzero only in 61% of the bootstraps, meaning that it was not stable. The boodstrapped differences of centrality scores, edge-weight accuracy, stability and differences are shown in the supplementary material.

**Table 1.** Demographic breakdown and treatment status (n = 2,796)

|  | Frequency | Percentage |
|---|---|---|
| Years of education |  |  |
| 0 | 1,157 | 41.4 |
| 1 to 9 | 1,009 | 36.1 |
| 10 to 14 | 307 | 11 |
| 15 to 17 | 35 | 1.3 |
| Missing | 288 | 10.3 |
| Employment |  |  |
| Unemployed | 1,664 | 59.5 |
| Part time | 406 | 14.5 |
| Full time | 374 | 13.4 |
| Student | 10 | 0.4 |
| Retired | 5 | 0.2 |
| Any other | 47 | 1.7 |
| Missing | 290 | 10.4 |
| Managing finances |  |  |
| Living comfortably | 254 | 9.1 |
| Just about getting by | 1,107 | 39.6 |
| Difficult to make the ends meet | 1,145 | 41 |
| Missing | 290 | 10.4 |
| Debt |  |  |
| Yes | 992 | 35.5 |
| No | 1,488 | 53.2 |
| Do not know | 26 | 0.9 |
| Missing | 290 | 10.4 |
| Treatment status at follow-up |  |  |
| Responders | 903 | 32.3 |
| Delayed remission | 297 | 10.6 |
| Relapse | 214 | 7.7 |
| Nonresponders | 443 | 15.8 |
| Missing | 939 | 33.6 |

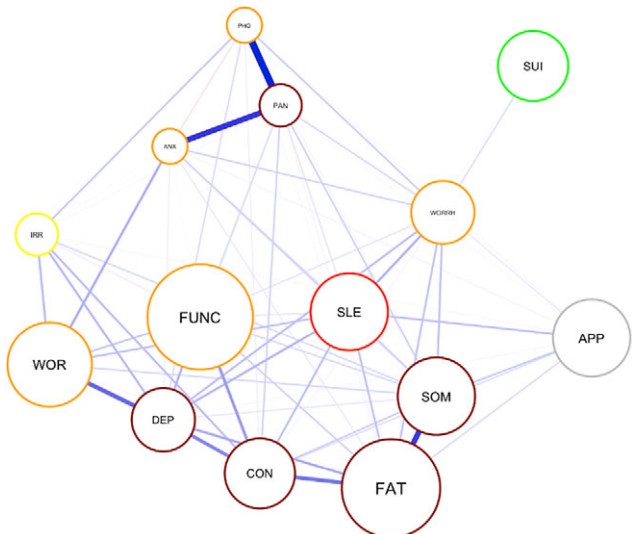

**Figure 1.** The Gaussian graph model (GGM) network of the 14 CMD symptoms Note: The size of the nodes represents the mean value. The colors represent node centrality in decreasing order; dark red, red, orange, yellow, green. *Note:* APP: appetite and weight changes, ARM: arm, ANX: anxiety, CON: concentration, DEP: depression, FAT: fatigue, FUNC: functional impairment, IRR: irritability, PAN: panic, PHO: phobia, SOM: somatic, SLE: sleep problems, SUI: depressive ideas, WOR: worry, WORRH: worry about health.

($p < 0.001$), while the edge between somatic-fatigue was significantly stronger in the private network ($p < 0.001$). The NCT does not compare node centralities across groups. Yet, the centrality indices were stable for both groups, meaning bootstrapped differences of centrality indices within each group are reliable. A visual inspection of the black boxes for centrality differences in each group separately (see Supplementary Material, Figures 9 and 14) revealed that in public settings, the strength centrality scores of panic and depressed mood were the highest centrality indices, while in private settings, it was fatigue, depressed mood, and somatic symptoms.

### Responders versus nonresponders (Aim 4)

The global strength of the networks from treatment responders ($n = 903$) and nonresponders ($n = 954$) were not significantly different ($p = 0.688$), with values of 2.203 and 2.556, respectively. The results were similar when the analyses were repeated separately for the intervention arm and control arm. No significant differences were found in the permutation test concerning the maximum difference in edge weights.

## Discussion

This study examined individual symptoms of CMDs in 2796 patients from public and private healthcare settings in India. Our goal was to investigate symptom-to-symptom relationships and diagnostic boundaries of CMDs in a non-Western country across diverse socioeconomic strata using network analysis. Results from the study indicate that, in line with most of the existing literature (e.g., Malgaroli et al. 2021), depressed mood and fatigue were among the most central CMD symptoms. More interestingly, across all models (including Ising and MGM, see additional models in the Supplementary Material), panic was at least as central as depressed mood and fatigue.

In line with our findings, panic was most central in a large online survey study using the DASS capturing symptoms of depression,

### Comorbidity networks (Aim 2)

The community detection using the walktrap random walk algorithm did not show strong evidence for a community structure; the modularity was only 0.043, too small to indicate partitioning (Newman and Girvan 2004).

### Public versus private healthcare settings (Aim 3)

The GGM was run for public and private healthcare patients (Figure 3). The CIS-R total scores, GHQ total scores, and the 12 CIS-R subscale scores across public and private settings are shown in Table 2. Strength centrality indices for both public and private settings were stable (CS coefficient = 0.75 for both groups). The network densities were not significantly different ($p = 0.157$), with descriptively higher values for public healthcare patients (4.73, $n = 1648$) than the private health care (4.34, $n = 1148$). The edges anxiety-panic and depression-worry were significantly stronger in the public network than the private healthcare network

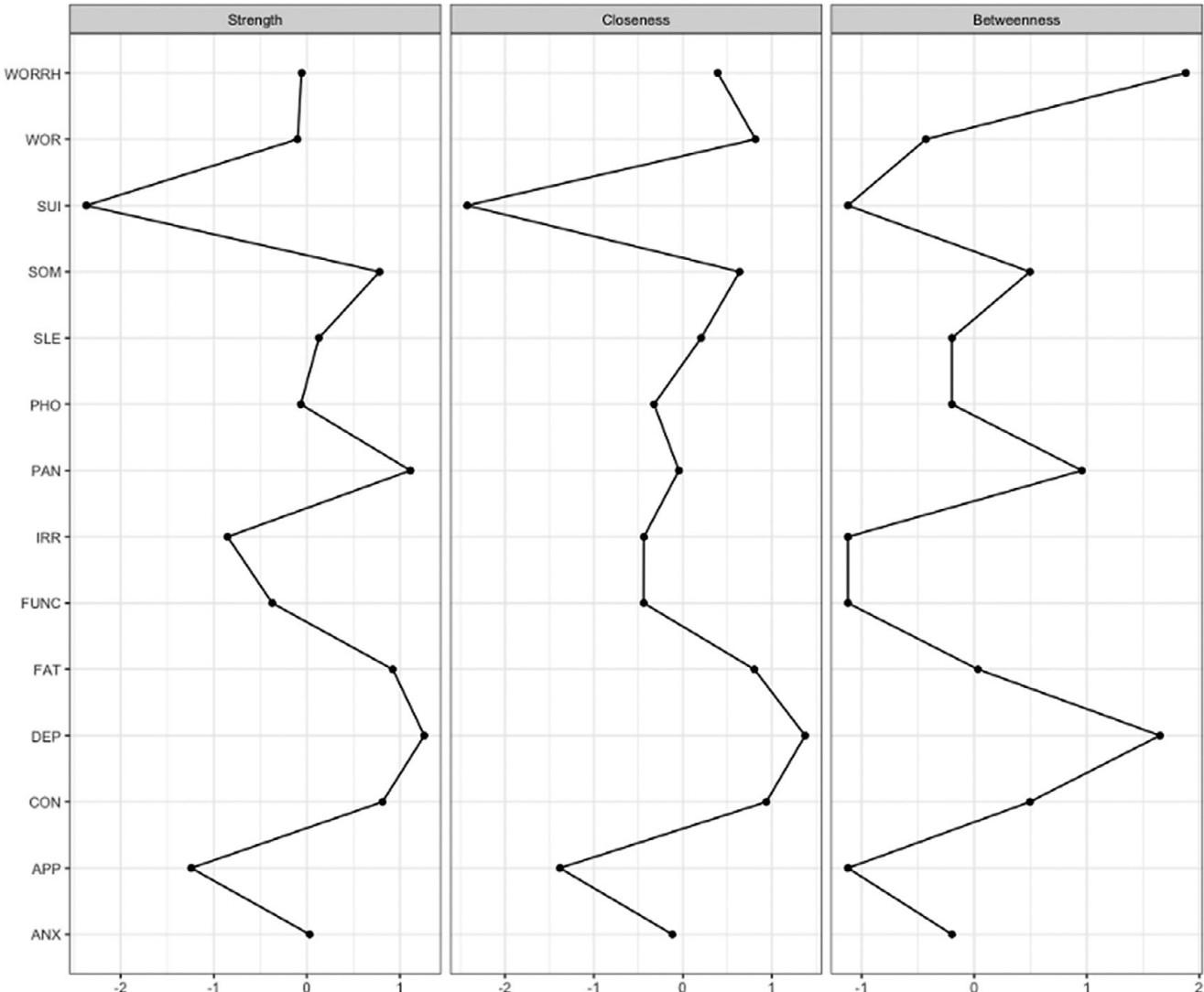

**Figure 2.** The centrality indices of the GGM network.

anxiety and stress in one single measure (Van den Bergh et al. 2021). While the participants of that study were predominantly from Western high-income-countries (US, UK, Canada and Australia altogether accounted for about 64% of the sample), a Southeast Asian middle-income country, Malaysia, was overrepresented (25% of the sample). Another study found clinician-assessed psychic anxiety of the HAM-D to be the most central symptom among clinically depressed adults in South Korea (Park and Kim 2020). Both studies used a unified measure to assess depression and anxiety symptoms, had female-majority samples (63% to 82%), and included Asian participants, either entirely or partially. It has been argued previously that typical pathways of depression might differ across men and women, such that an anxiety-pathway could be more plausible for women (Kessler 2003). Similarly, different pathways might be at play across different cultures or income levels.

These findings are corroborated by qualitative studies conducted in the region. In an ethnographic study, the authors reported that "tension" as a local idiom could be a central feature of psychological distress in South Asia, with the most common features being anger, worry, nervousness or restlessness

(Weaver 2017). In fact, in a previous qualitative study nested within the same MANAS trial (Patel et al. 2010), half of the 117 primary care patients named their health problem as "tension" (Andrew et al. 2012). In CIS-R, panic was conceptualized similar to a sense of extreme anxiety or tension: "Thinking about the past month, did your anxiety or tension ever get so bad that you got in a panic, for instance, make you feel that you might collapse or lose control unless you did something about it?" Our findings indicate intense anxiety or panic might play an important role in illness development in some groups.

Panic/extreme anxiety is differentiated from other symptoms (such as worry, worry about health) through its ties with sympathetic activation, potentially indicating a threat/stress response. Thus, this finding could be about the context-specific presentation of the CMDs, changing across different levels of perceived environmental threat, one source possibly being financial distress. In fact, the centrality score of panic might be driven by public health-care attendees who constitute the majority in this sample. Health care in India is heavily privatized and related expenditures are a leading cause of poverty (Reddy et al. 2011). Only those with higher resources (both assets and social networks) could access private

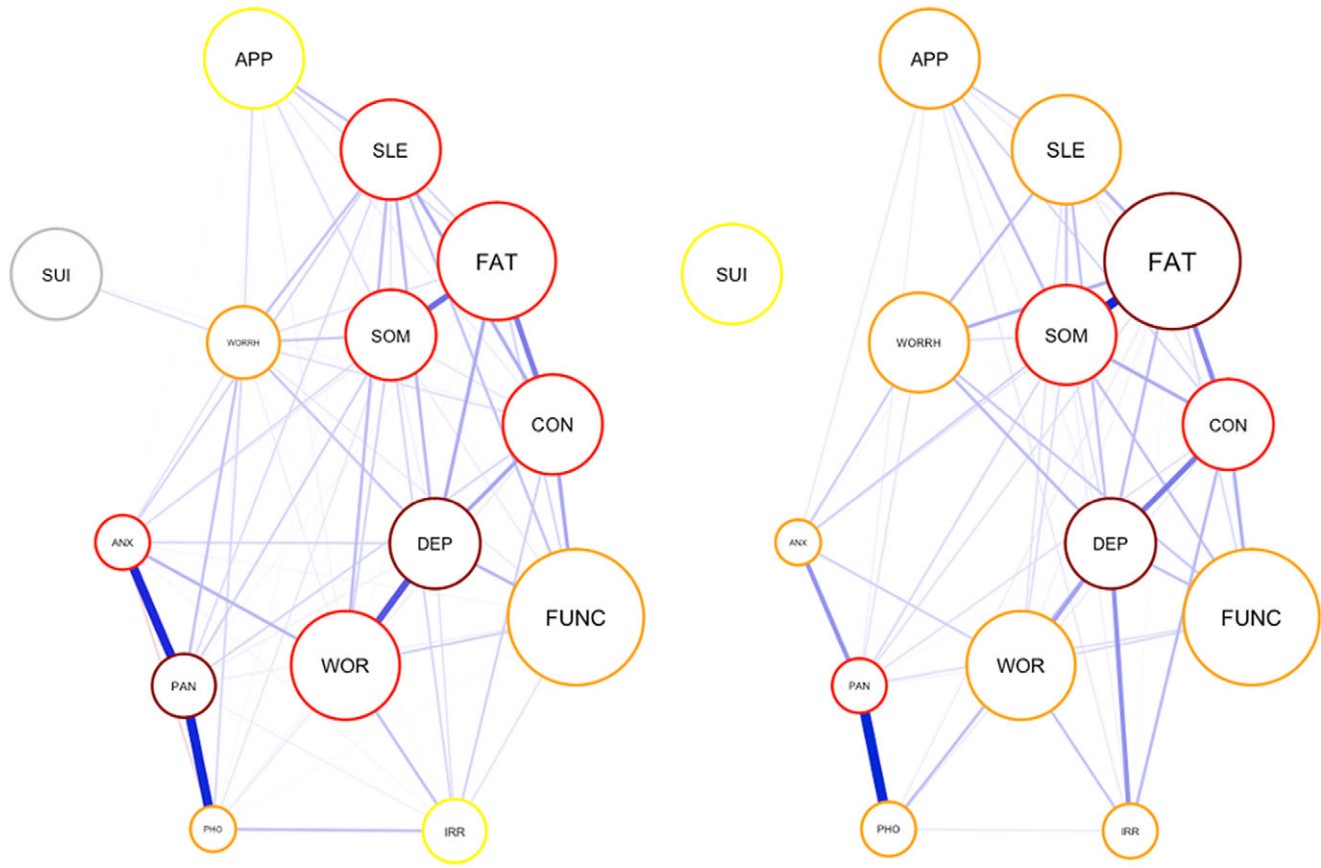

**Figure 3.** The GGM of the 14 symptoms across public ( *n* = 1648, on the left) and private ( *n* = 1148, on the right) settings. Note: The size of the nodes represent the mean value of the node. The colors represent node centrality in the following decreasing order; dark red, red, orange, yellow, green. *Notes:* APP: appetite and weight changes, ARM: arm, ANX: anxiety, CON: concentration, DEP: depression, FAT: fatigue, FUNC: functional impairment, IRR: irritability, PAN: panic, PHO: phobia, SOM: somatic, SLE: sleep problems, SUI: depressive ideas, WOR: worry, WORRH: worry about health.

**Table 2.** The 14 subscale scores across PHC and GP

| | PHC | | GP | | Total | |
| --- | --- | --- | --- | --- | --- | --- |
| | (n = 1,648) | | (n = 1,148) | | (n = 2,796) | |
| | Mean | SD | Mean | SD | Mean | SD |
| Somatic | 1.82 | 1.65 | 1.97 | 1.58 | 1.88 | 1.63 |
| Fatigue | 3.21 | 1.65 | 3.34 | 1.66 | 3.27 | 1.66 |
| Concentration | 1.87 | 1.27 | 1.71 | 1.34 | 1.81 | 1.3 |
| Sleep | 1.95 | 1.33 | 2.07 | 1.27 | 0.93 | 1.17 |
| Irritability | 0.97 | 1.21 | 0.86 | 1.11 | 2 | 1.3 |
| Worry health | 1.34 | 1.5 | 1.96 | 1.35 | 1.6 | 1.47 |
| Dep. mood | 1.63 | 1.21 | 1.62 | 1.22 | 1.62 | 1.21 |
| Worry | 2.22 | 1.62 | 2.02 | 1.54 | 2.14 | 1.59 |
| Dep. ideas | 2.17 | 1.56 | 2.34 | 1.54 | 2.24 | 1.56 |
| Anxiety | 0.69 | 1.2 | 0.62 | 1.04 | 0.66 | 1.14 |
| Phobias | 0.52 | 1.02 | 0.79 | 1.21 | 0.86 | 1.5 |
| Panic | 0.92 | 1.58 | 0.78 | 1.35 | 0.63 | 1.11 |
| Appetite | 0.94 | 0.74 | 0.99 | 0.8 | 0.96 | 0.76 |
| Functioning | 2.21 | 0.91 | 2.07 | 1 | 2.15 | 0.95 |

PHC = public health care.
GP = general practitioner (private health care).

health care. Thus, one hypothesis generated from this study is, for primary care patients who suffer from the lack of healthcare resources, a sense of panic is more quickly activating the rest of the network (or vice versa) compared to those who have sufficient resources, and for that particular group, this is the mechanism through which the system is shifting from a healthy to an unhealthy state.

The density of the responder network, although descriptively weaker, was not significantly different than the network of those who did not show sustained response. This might be due to the heterogeneity of the nonresponse category. For instance, those who responded late (at 6 or 12 months) could have been more similar to the responders group, thereby serving as a confounder. Yet, the lack of significant difference is in line with others (Schweren et al. 2018; O'Driscoll et al. 2021) and might indicate that in the treatment context, strong edges could in fact serve some patients who began experiencing improvements, by starting a favorable activation in the overall network.

The current study found little evidence to support the notion that anxiety and depression are distinctive conditions. The modularity index being too small to indicate distinct communities for anxiety and depression-specific symptom subscales, there was more evidence for the unidimensionality of the CMDs measured by the CIS-R. This might support the idea that these conditions could not be separated from each other clearly, when a nondiagnostic approach is taken, particularly among primary care attendees in a non-Western, primary care setting.

### Strengths, limitations and implications for future research

The study had two major methodological strengths. First, the sample consisted of all primary care patients who screened positive on a health questionnaire as opposed to only depressed or anxious participants as it has typically been done. If we had included only diagnosed patients, this would have limited our ability to question the diagnostic categorization, reduced variability and potentially led to erroneous network estimation (De Ron et al. 2021). Second, the assessment had a symptom-focused approach rather than a diagnostic focus, where all culturally relevant symptoms were assessed, and they were assessed only once with more than one item, in a structured interview.

Some methodological challenges existed. First, a decision needed to be made about how to treat the variables given the skewed data distribution. This article only presents the findings when the variables are treated as continuous, though different models are also examined treating the variables as binary and mixed of binary and continuous. Second, this study might have missed important symptoms and/or have included topologically similar symptoms. The use of composite scores might have overshadowed the importance of some symptoms such as guilt and self-blame. Third, while the notion of centrality seems intuitive, its predictive value is unclear. A symptom might be central because it is the "causal endpoint" of a pathway, in which case, intervening on that symptom may not lead to changes in the system (Fried et al. 2018). Strong edges are potentially loaded with information pertaining to the mechanism of change. Finding a strong edge might be a good start though will offer limited clinical implications without knowing the causal mechanism and other factors behind it. Important external factors (latent or observed), however, might contribute to the links between symptoms, including weakened social support, adaptive coping skills, genetic predisposition and/or neural correlates. Thus, experimental data and longitudinal within-person designs are required to infer a causal chain between symptoms. However, even such research design is not without limitation: isolating one target symptom for intervention is almost impossible clinically, previously referred to as the "fat-hand" problem (Bringmann et al. 2019), where a psychosocial intervention might cause changes in more than one symptom.

To move beyond these limitations, a suggestion for future research is to focus on the "network as a whole" (Bringmann et al. 2019) rather than centrality alone. The "network connectivity/density" (i.e., the global strength) and identification of symptom hubs might have better prognostic value (Cramer et al. 2016). With that perspective, the fact that no specific communities were found in our analyses suggest that all CIS-R symptoms could play an equal role in illness development.

This study examined the CMD symptom networks in a South Asian low-income country and explored the differences in network constellation across different treatment response groups and socio-economic backgrounds. The findings highlighted the importance of intense anxiety/panic particularly among public healthcare patients. If this is true, a major public healthcare implication might be about screening for panic symptom, or "intense anxiety/tension" in primary care settings, to identify those that are at risk of developing or have developed a CMD. However, future studies investigating causality through repeated measures or experimental designs are required before any public health recommendations could be made. Rather than exclusively focusing on the "centrality" of individual symptoms, examining network density, identifying clusters of symptoms with strong reciprocal relationships, and the mechanisms through which symptoms exacerbate each other (i.e., unpacking the edges) is recommended for future studies.

**Open peer review.** To view the open peer review materials for this article, please visit http://doi.org/10.1017/gmh.2025.16.

**Supplementary material.** The supplementary material for this article can be found at http://doi.org/10.1017/gmh.2025.16.

**Data availability statement.** Data originated from the MANAS Cluster Randomized Controlled Trial. Requests for access to these data should be made to Vikram Patel (Vikram_Patel@hms.harvard.edu).

**Author contribution.** All authors contributed to the conceptualization, methodology, data analysis and interpretations of the results. First author wrote the original first draft and all authors contributed by reviewing, commenting and editing the subsequent versions.

**Financial support.** This research did not receive any specific grants from funding agencies in the public, commercial or not-for-profit sectors. MM was supported by the National Institute of Mental Health (NIMH) award K23MH134068. The content is solely the responsibility of the authors and does not necessarily represent the official views of the NIMH.

**Competing interest.** The authors have no affiliation with any organization with a direct or indirect financial interest in the subject matter discussed in the manuscript. This manuscript has not been submitted to, nor is under review at, another journal or other publishing venue.

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
