## [Reviewer Report]

This paper examines a partial correlation network of symptoms of common mental disorders (CMD) in India. The authors provide a solid description of methods and their rationale for these methods, including statistical approaches and measurement selection. A few comments for consideration are included below:

(1) In the introduction, second paragraph, it would be helpful to have a citation for the common cause model.

(2) In the second paragraph, a citation about central nodes serving as good therapy targets would be help.

(3) Great summary of methodological limitations of comorbidity networks and how the authors circumvent these concerns with their measure selection.

(4) In the present study section, what did participants screen positive for? (“In the MANAS trial, researchers included all primary care participants who screen positive on a health questionnaire.” line 162)

<b>(5) In the present study section, under aim 2, line 179, there is a typo whereby the authors refer to the fourth aim during the aim 2 section. Similarly, under aim 3, line 197, the authors refer to the second aim. Lastly, the authors refer to the third aim during the fourth aim section, line 212. </b>

(6) In the methods section, the authors state that “only baseline data are used to construct the symptom networks for the present study”; however, aim 4 appears to include various timepoints.

(7) Please include citations when describing centrality metrics in lines 257-262.

(8) The authors describe the network estimation process very well

(9) Please include the treatment that was received for CMD.

(10) It would be helpful if the authors included the centrality scores for the various nodes.

(11) Please differentiate between (1) depressed mood and depressive ideas and (2) worry and anxiety.

(12) In the discussion, line 524, the authors state there is little evidence to support anxiety and depression as distinctive conditions. It would be helpful if the authors could re-iterate the evidence here (e.g., no evidence of diagnosis-specific clustering/ communities)

---

## [Editor Report]

This is a very well-written paper. Rationale, methodology, and results are described well. The last sentence of the introduction mentions, ‘Thus, the third aim of the present study is to compare the symptom network density of treatment responders at 2, 6, and 12-month follow-up, versus non-responders’, while the section related to participants mentions, ‘only baseline data are used to construct the symptom networks for the present study.’ Kindly clarify.